# The Potential Effects of Quercetin-Loaded Nanoliposomes on Amoxicillin/Clavulanate-Induced Hepatic Damage: Targeting the SIRT1/Nrf2/NF-κB Signaling Pathway and Microbiota Modulation

**DOI:** 10.3390/antiox12081487

**Published:** 2023-07-25

**Authors:** Mahran Mohamed Abd El-Emam, Mahmoud Mostafa, Amina A. Farag, Heba S. Youssef, Azza S. El-Demerdash, Heba Bayoumi, Mohammed A. Gebba, Sawsan M. El-Halawani, Abdulrahman M. Saleh, Amira M. Badr, Shorouk El Sayed

**Affiliations:** 1Department of Biochemistry and Molecular Biology, Faculty of Veterinary Medicine, Zagazig University, Zagazig 44511, Egypt; mmmahran@vet.zu.edu.eg; 2Department of Pharmaceutics, Faculty of Pharmacy, Minia University, Minia 61519, Egypt; mahmoud_mohamed@mu.edu.eg; 3Department of Forensic Medicine and Clinical Toxicology, Faculty of Medicine, Benha University, Banha 13518, Egypt; amina.farag@fmed.bu.edu.eg; 4Department of Physiology, Faculty of Medicine, Benha University, Benha 13518, Egypt; heba.youssef@fmed.bu.edu.eg; 5Laboratory of Biotechnology, Department of Microbiology, Agriculture Research Centre (ARC), Animal Health Research Institute (AHRI), Zagazig 44516, Egypt; dr.azzasalah@yahoo.com; 6Department of Histology and Cell Biology, Faculty of Medicine, Benha University, Benha 13518, Egypt; heba.bayoumi@fmed.bu.edu.eg; 7Department of Anatomy and Embryology, Faculty of Medicine, Benha University, Benha 13518, Egypt; mohammedgebba@gmail.com; 8Department of Anatomy and Embryology, Faculty of Medicine, Merit University, Sohag 82524, Egypt; 9Department of Biotechnology, Urology and Nephrology Center, Mansoura University, Mansoura 35516, Egypt; sawsanelhalawani@mans.edu.eg; 10Pharmaceutical Medicinal Chemistry & Drug Design Department, Faculty of Pharmacy (Boys), Al-Azhar University, Cairo 11884, Egypt; abdo.saleh240@azhar.edu.eg; 11Pharmacology and Toxicology Department, Faculty of Pharmacy, King Saud University, Riyadh P.O. Box 11451, Saudi Arabia; 12Department of Microbiology, Faculty of Veterinary Medicine, Zagazig University, Zagazig 44511, Egypt; shorokelsaid@zu.edu.eg

**Keywords:** Co-Amox-induced hepatotoxicity, quercetin liposomes, quercetin antioxidant activity, SIRT1, Nrf2 and NF-κB targeting, gut dysbiosis

## Abstract

Amoxicillin/clavulanate (Co-Amox), a commonly used antibiotic for the treatment of bacterial infections, has been associated with drug-induced liver damage. Quercetin (QR), a naturally occurring flavonoid with pleiotropic biological activities, has poor water solubility and low bioavailability. The objective of this work was to produce a more bioavailable formulation of QR (liposomes) and to determine the effect of its intraperitoneal pretreatment on the amelioration of Co-Amox-induced liver damage in male rats. Four groups of rats were defined: control, QR liposomes (QR-lipo), Co-Amox, and Co-Amox and QR-lipo. Liver injury severity in rats was evaluated for all groups through measurement of serum liver enzymes, liver antioxidant status, proinflammatory mediators, and microbiota modulation. The results revealed that QR-lipo reduced the severity of Co-Amox-induced hepatic damage in rats, as indicated by a reduction in serum liver enzymes and total liver antioxidant capacity. In addition, QR-lipo upregulated antioxidant transcription factors SIRT1 and Nrf2 and downregulated liver proinflammatory signatures, including IL-6, IL-1β, TNF-α, NF-κB, and iNOS, with upregulation in the anti-inflammatory one, IL10. QR-lipo also prevented Co-Amox-induced gut dysbiosis by favoring the colonization of *Lactobacillus*, *Bifidobacterium*, and *Bacteroides* over *Clostridium* and *Enterobacteriaceae*. These results suggested that QR-lipo ameliorates Co-Amox-induced liver damage by targeting SIRT1/Nrf2/NF-κB and modulating the microbiota.

## 1. Introduction

The liver is a key organ in the process of drug metabolism. In clinical settings, drug-induced liver injury (DILI) is often reported by physicians [1]. Moreover, DILI is caused by exposure to pharmaceuticals, herbal remedies, or other xenobiotics. Antibiotics are frequently accused of causing autoimmune hepatitis, drug-induced liver destruction, and liver failure following transplantation [2]. Research has shown that amoxicillin-clavulanate (co-amoxiclav (Co-Amox)) is the causative agent that is often associated with DILI [3]. Co-Amox is considered a broad-spectrum antibiotic combination composed of the antibiotic amoxicillin, a semimanufactured antibiotic, and an inhibitor for the enzyme β-lactamase called potassium clavulanate. Many drug combinations of amoxicillin and clavulanate have been released globally over time to improve the ease of dosing, the requirements for prescribing, and the recommended treatments for more serious infections or those caused by bacterial resistance to antibiotics [4]. Co-Amox is commonly prescribed for bacterial infections, including sinusitis, otitis, bacterial bronchitis, and pneumonia [5]. Several investigations have indicated that Co-Amox-induced hepatotoxicity may occur, despite the drug’s designated conservation aims [6]. The chance of liver damage and hepatotoxicity rises when amoxicillin is administered with a beta-lactamase inhibitor, such as clavulanic acid [7]. In addition, most DILI-related hospitalizations in clinical medicine are caused by Co-Amox. Co-Amox-induced liver damage could be cholestatic, hepatocellular, and/or mixed damage with hypersensitivity symptoms in certain circumstances [8].

Furthermore, evidence suggests that reactive oxygen species (ROS)-induced oxidative damage is crucial in the pathophysiology of Co-Amox-induced hepatocellular injury. When the endogenous antioxidant system is depleted, free radical scavengers become insufficient, resulting in the initiation of negative consequences [8]. Sirtuin1 (SIRT1) is considered a significant factor in DILI development and can influence a variety of biological activities and processes by controlling specific crucial signaling pathways in antistress, autophagy, and DNA repair. It has been suggested that SIRT1, which is a class III histone deacetylase, has protective effects on hepatocytes since it is routinely expressed in hepatic tissue [9]. However, it is downregulated as an effect of hepatocellular damage. Likewise, the reduced hepatic SIRT1 could aggravate DILI primarily by decreasing the nuclear factor E2-related factor 2 (Nrf2), which significantly boosts the expression of antioxidant enzymes [10].

Substantial ROS production can also induce numerous inflammatory factors that trigger inflammatory responses in the liver. The NF-κB signaling pathway plays a major role in the understanding of the mechanism of inflammation [10]. NF-κB is a dimer protein that aids in the production of various proinflammatory cytokines and induces inflammation as a result [11]. Consequently, liver disease prevention and protection are essential. Interest in the potential applications of natural antioxidants as medicinal agents and immune stimulants has been growing [12,13]. Therefore, increasing antioxidant and anti-inflammatory potency and targeting SIRT1 may mitigate DILI by reducing oxidative stress and promoting tissue regeneration.

Microbiota represent a whole organism as they consist of trillions of microbes, composed of different species with diverse taxonomies in humans, including bacteria and fungi [14]. Regulation of intestinal microbiota ecology is multifactorial through microbial, host immune response, environment interaction, and genetic susceptibility. Disturbance in this interaction in genetically susceptible individuals would lead to microbiota dysbiosis [14]. Recently, the role of microbiota dysbiosis not only in the induction of liver disease but also in the fate of disease pathology and therapy has been reported [14,15]. Patients with coeliac disease have been known to develop liver problems, such as nonalcoholic fatty liver disease (NAFLD) and primary biliary cirrhosis [16]. Moreover, the translocation of buccal microbiota to the gut has been documented in individuals with liver cirrhosis whose innate immune surveillance is impaired [17]. The overuse of antibiotics leads to microbiota dysbiosis and accelerates the evolution of antibiotic-resistant bacteria [18]. Microbiota dysbiosis affects barrier function, microbiota diversity, and metabolite products, and subsequently impairs the efficiency of hepatoprotective substances [16]. Therefore, antibiotics are important in the progression of liver disease pathogenicity. In this respect, introducing natural prebiotics or probiotic substances to be used either as supplements or antibiotic alternatives to modulate gut microbiota composition is an alternative [19,20]. Previous studies have addressed the role of natural flavonoids, including QR, in modulating microbiota and their metabolite composition [21,22,23].

Quercetin (QR) is a plant flavonoid that is found in a variety of vegetables, fruits, and seeds, including broccoli, onions, soybeans, and peanuts, as well as drinks created from plants, such as tea and wine [24]. Due to its pleiotropic biological properties, including its anti-inflammatory and antioxidant properties, QR has gained more and more attention. QR has been demonstrated to possess pharmacologically proven neuroprotective [25], cardioprotective [26], hepatoprotective, and nephroprotective properties [27]. Furthermore, its potential use in clinical medicine has also recently been reviewed. Additionally, QR is a prospective liver defender since it can directly neutralize superoxide anion, inhibit various superoxide-producing enzymes, including xanthine oxidase, and maintain levels of reduced glutathione [28]. However, despite its medical potential, the clinical use of QR is severely constrained because of its limited bioavailability and reduced water solubility [29]. Therefore, it is crucial to incorporate QR into drug delivery systems that may enhance its bioavailability. Liposomes, a well-known lipid-based drug delivery system, have good biocompatibility, high drug-loading efficiency, controlled release properties, and can potentially encapsulate lipid-soluble [30] and water-soluble [31] molecules. As a result, drugs that are poorly water-soluble, including QR, are more bioavailable upon formulation into liposomal systems [32]. QR might induce hepatoprotection through activation of SIRT1, which is a potent anti-inflammatory factor [33]. The cocrystallized ligand, resveratrol, that binds to the Sirt1 crystal protein (PDB ID: 5BTR), was utilized as a reference to evaluate the potential of QR to bind SIRT1 via docking [34]. QR can imitate resveratrol’s action by binding to the activator region of the SIRT1 protein, therefore activating the SIRT1 pathway [35].

However, to our knowledge, there are no reports of the ameliorative effects of QR nanoliposomes on Co-Amox-induced liver damage in male rats in a SIRT1/Nrf2/NF-*κ*B pathway or gut-liver axis-dependent manner. In light of this, the current study was conducted to assess the protective characteristics of QR in the form of liposomes and to identify the potential mechanisms of action.

## 2. Materials and Methods

### 2.1. Chemicals

Augmentin^®^ (amoxicillin/clavulanate potassium) powder for suspension (Glaxo Smith Kline, Brantford, UK) was purchased from a public pharmacy in Egypt. Quercetin (95% purity) and cholesterol were provided by Sigma–Aldrich (St. Louis, MO, USA). Phospholipid 90 G was purchased from Lipoid GmbH (Ludwigshafen, Germany). All other chemicals, solvents, and reagents were of the highest purity.

### 2.2. Animals

Twenty-eight male Sprague-Dawley rats weighing between 160 and 170 g were purchased from the experimental animal’s unit of the College of Veterinary Medicine at Zagazig University (Zagazig, Egypt) for this work. The rats were provided with regular laboratory commercial feed and water before the experiment, and acclimatized for two weeks at approximately 25 °C.

### 2.3. Preparation of a Nanoliposomal Formulation for Quercetin

Quercetin liposomes (QR-lipo) were developed according to the previously disclosed solvent injection approach [36,37]**.** In brief, Phosopholipon 90 G (15.5 mg/mL), cholesterol (1.5 mg/mL), and QR (3 mg/mL) were dissolved in a sufficient volume of absolute ethanol, forming the organic phase. The organic phase was kept under heat in a closed system until used. Ten milliliters (mL) of deionized water were kept under stirring (750 rpm) at 60–70 °C to form the aqueous phase. The organic solution was injected into the aqueous phase using a syringe (25 G needle). QR-lipo production was indicated by the aqueous medium turning into a milky mixture during the injection process. The resulting suspension was maintained at 60 °C for 20–30 min to facilitate ethanol evaporation. The mixture was then stirred continuously for 1–2 h at room temperature and finally stored at 4 °C until further characterization. The resultant liposomal suspension phospholipid concentration is 20 mM containing 20 mM% cholesterol. Encapsulation efficiency (%) of QR was estimated spectrophotometrically (UV-1900, Shimadzu, Kyoto, Japan) at 373 nm after the lysis of liposomal systems as previously described [38]. QR-lipo was evaluated for several parameters, including particle mean diameter, particle charge, polydispersity index, and transmission electron microscopy. The in vitro release profile of QR in the form of liposomes was also examined, as previously reported [30]. An aliquot of QR-lipo was placed in the sample compartment of a Franz diffusion cell. Water and ethanol were mixed in a ratio of 65:35 in the release medium and placed in the reservoir compartment to maintain sink conditions that replicate the in vivo environment [39]. A nitrocellulose membrane with a molecular weight cutoff of 12 to 14 KDa separated the two compartments. System temperature and speed were maintained at 37 °C and 50–60 rpm, respectively. Two mL of the dissolving medium were collected at regular intervals and put through UV analysis at 373 nm to quantify QR using a standard curve. An equal volume of release medium maintained at 37 °C was introduced in the reservoir compartment.

### 2.4. Experimental Design

Rats were randomly allocated into four groups (*n* = 7), control group, quercetin liposome-treated group (QR-lipo), the Co-Amox treated group, and the Co-Amox group treated with QR-lipo (Co-Amox and QR-lipo). The liver injury was induced by Co-Amox oral suspension at a dose of 60 mg/kg for ten consecutive days [8]. Rats in the Co-Amox and QR-lipo group received daily doses of 5 mg/kg of QR-lipo intraperitoneally for 10 consecutive days, 1 h before receiving an oral suspension of Co-Amox [37]. Rats in the control group were only given saline solution. Rats in the QR-lipo treatment received daily intraperitoneal injections of 5 mg/kg of QR-lipo for ten days without ingesting Co-Amox. Figure 1 is a schematic diagram summarizing the experimental design.

### 2.5. Sampling

The rats were sacrificed by carbon dioxide exposure and necropsied 24 h after the last co-amoxiclav treatment. Blood samples were immediately obtained from caudal vena cava before necropsy. After that, the samples used for biochemical research were stored at room temperature in order to isolate serum devoid of anticoagulants. The liver was removed and divided into three parts. After homogenizing a portion in ice-cold phosphate-buffered saline (PBS), the supernatant was utilized to determine the antioxidant status. The second portion was utilized to analyze gene expression using real-time PCR. The remaining liver portion was stored in a 10-percent buffered neutral formalin solution for histological and immunohistochemical analysis.

### 2.6. Biochemical Analysis

Alanine transaminase (ALT), aspartate transaminase (AST), and total albumin levels were determined by the colorimetric method according to kits supplied by Chema Diagnostica (Monsano, Italy), Tulip Diagnostics (Chennai, India), and Agape Diagnostics (Cham, Switzerland), respectively.

### 2.7. Determination of the Antioxidant Status

The sample’s antioxidants were mixed with a predetermined amount of exogenously generated hydrogen peroxide (H_2_O_2_) to calculate the liver supernatant’s overall antioxidant capacity. The antioxidants in the sample neutralize some of the provided hydrogen peroxide, while the residual H_2_O_2_ concentration was determined using a calorimetric enzymatic reaction that involved the production of a colored product from 3,5-dichloro-2-hydroxybenzenesulfonate. The total antioxidant level was calculated using the following equation [40]:(Absorbance of blank) – (absorbance of the sample) × 3.33 (mM/L)(1)

Catalase (CAT) activity was estimated calorimetrically in the supernatant by the method described by Hugo Aebi et al. [40]. A catalase inhibitor is used to stop the reaction between a known quantity of H_2_O_2_ and CAT after exactly one minute. Horseradish peroxidase allowed 3,5-dichloro-2-hydroxybenzene sulfonic acid and 4-aminophenazone to react with the remaining H_2_O_2_ to produce a chromophore with a color intensity that was inversely correlated to the amount of catalase in the initial sample [40]. Malondialdehyde (MDA) level in liver tissue was determined by the method described by Hiroshi Ohkawa et al. [41]. A thiobarbituric acid (TBA) reactive product was created when TBA and MDA reacted at 95 °C for 30 min in an acidic medium to form a colored product that could be estimated spectrophotometrically at 534 nm. Reduced glutathione (GSH) levels in liver tissue were estimated by the reduction of 5,5-dithiobis (2-nitrobenzoic acid) (DTNB) with GSH to form a yellow-colored chromogen [42]. The absorbance of the resulting chromogen at 405 nm was determined using a commercial kit (Bio Diagnostic, Cairo, Egypt), and it was directly proportional to GSH content.

### 2.8. mRNA Quantification Using Real-Time RT-PCR

The QIAamp RNeasy Mini kit was used to extract and purify the total RNA from liver tissue in accordance with the manufacturer’s instructions (Qiagen, Hilden, Germany). Thermo Fisher Scientific’s Nanodrop 8000 was used to assess the amount of total RNA present. A list of the primers used in this work was provided by Metabion (Planegg, Germany), as shown in Table 1. The reaction was carried out in a 25 µL running volume using 10 µL of the 2× HERA SYBR^®^ Green RT-qPCR Master Mix (Willow Fort, Nottinghamshire, UK), 1 µL of the RT Enzyme Mix (20×), 0.5 µL of each primer at a concentration of 20 pmol, 5 µL of RNAse-free water, and 3 µL of RNA template. A step-one real-time PCR instrument was used to conduct the experiment. The reverse transcription procedure was carried out at 50 °C for 30 min, the cDNA was denatured at 94 °C for 15 min, and then 40 cycles of 95 °C for 15 s and 60 °C for 30 s were used in the PCR to achieve amplification. According to the approach outlined by Yuan et al.; [43], in order to evaluate the variation in gene expression on the RNA of the different samples, the Ct of each sample was calculated using the 2^–ΔΔCt^ method and normalized to those of β-actin as the housekeeping gene.

### 2.9. Real-Time Quantitative of Bacterial Population Abundance in Cecal Contents

The bacterial DNA was extracted from cecal contents of all rat groups for detection of microbial abundance of the following bacterial species, including *Lactobacillus, Bacteroides*, *Enterobacteriaceae*, *Bifidobacterium,* and *Clostridium,* using Strata gene MX3005P quantitative real-time PCR (RT-PCR). Using QIAamp Fast DNA Stool Mini (Qiagen, Hilden, Germany), total DNA from a sample of cecal material was extracted. Using a Nanodrop 2000 spectrophotometer (ThermoFisher Scientific Inc.; Waltham, MA, USA), the purity and concentration of extracted DNA were measured. The samples were frozen at −80 °C for subsequent quantitative PCR analysis. The sequence of the primers for selected bacterial species is mentioned in Table 1. Triplicate analyses for PCR amplification were conducted in a 25 μL reaction containing the following mix: 1 μL of each primer (10 mM), SYBR Green PCR Master Mix (12.5 μL) (Qiagen, Hilden, Germany), sterile PCR-grade water (9.5 μL) and specific genomic DNA (2 μL). For the development of standard curves, genomic DNA obtained from pure bacterial cultures was serially diluted tenfold. The Ct threshold cycle values were then plotted against the bacterial DNA copy counts to generate standard calibration curves. The standard curves represented log^10^ CFU/gram of the fecal contents and quantified the bacterial concentrations in each DNA sample.

### 2.10. Histological Examination of the Rat Liver

After obtaining a sample of rat liver, it was promptly fixed in a 10% buffered neutral formalin solution for 48 h, dehydrated in progressively increasing alcohol concentrations, cleaned in xylene, and then embedded in paraffin. Five-µm-thick paraffin slices were generated using a microtome (Leica RM 2155, Milton Keynes, England), and dewaxed sections were stained with hematoxylin and eosin (H and E) [56]. Finally, Leica^®^ microscope and an Am Scope^®^ microscope digital camera were used to take all section photos. The following evaluations of the lesions scoring system were made: 0 = no discernible histological changes, 1 = infrequently mild or focal, 2 = multifocal, and 3 = patchy or diffuse) using a semiquantitative approach [57].

### 2.11. Immunohistochemical Staining

Deparaffinized 5-μm-thick tissue slices were incubated with 3% H_2_O_2_ for 30 min, after which they were incubated for 1 h at 37 °C with anti-Nrf2 (GTX103322, Genetex, Alton Pkwy Irvine, CA, USA, 1:100) and anti-SIRT1 (ab110304, Abcam, Waltham, MA, USA, 1:70) reagents, following the manufacturer’s instructions. Cross-sections were treated with the secondary antibody HRP Envision kit (DAKO) for 20 min after being rinsed with PBS. The slices were then washed with PBS and given a 10-min incubation with diaminobenzidine (DAB). They were then dehydrated, cleaned in xylene, counterstained with hematoxylin, and cover slipped for microscopic analysis. The analysis was completed using the technique adopted from Elsayed et al. [58]. Seven representative nonoverlapping fields were randomly selected and scanned in order to determine the relative mean Area (%) of positive immunohistochemistry expression levels of Nrf2 and SIRT1 in immune-stained sections for each tissue section per sample. Data were gathered for the investigation of tissue sections utilizing a Full HD microscopic imaging system (Leica Microsystems Ltd.; Wetzlar, Germany) run using Leica Application software version 3.7.5 for tissue section analysis.

### 2.12. Molecular Docking Analysis

Using computer-based chemistry approaches and Resveratrol as a reference (Co-crystallized ligand), the probable activity of QR against the SIRT-1 target site was investigated. At first, the target protein was downloaded from the protein data bank (protein Id: 5BTR). All proteins, QR, and resveratrol were prepared, and an MMFF94 force field minimized energy. The key amino acids (Lys444, Asp292, Ala295, Asp298, and Pro212), which are responsible for the activation of the SIRT1 domain by activator attachment (resveratrol) were identified [34]. After completing the molecular docking, 20 positions were generated. The optimal orientations were then obtained, and affinity scores and RMSD values were compiled [59].

### 2.13. Statistical Analysis

All the graphs were performed using GraphPad Prism software version 9.5.1 (GraphPad Software, San Diego, CA, USA). One-way ANOVA followed by Tukey-Kramer Multiple comparison test was used to determine the statistical significance of the comparison between experimental groups. The significance levels were set at ns (nonsignificant) *p* > 0.05, * *p* < 0.05, ** *p* < 0.01, and *** *p* < 0.001.

## 3. Results

### 3.1. Characterization of the Liposomal Formulation of Quercetin

We first prepared QR-lipo via the ethanol injection method and characterized it for particle size, polydispersity index, TEM, and an in vitro release study. QR-lipo’s average particle diameter was 216.2 ± 15.7 nm, falling inside the nanometer range (Figure 2a). With a polydispersity value of 0.231, the liposomal suspension also demonstrated more uniform particle dispersion. Additionally, liposomal systems have a marginally negative surface charge, with a zeta potential of −10.3 ± 3.83 mV (Figure 2b). The spherical nature of the QR-lipo was demonstrated in the images captured by transmission electron microscopy (Jeol JEM-1010, Tokyo, Japan) with an average particle diameter similar to that determined using Zetasizer nano (Malvern, UK) (Figure 2c).

Furthermore, the liposomal systems demonstrated a better capacity to entrap QR, with an entrapment efficiency of roughly 71%. QR’s cumulative release rate from the liposomal dispersion is shown in the in vitro release testing (Figure 2d). In the first six hours, 40% of the QRs that had been loaded were released; following that, this percentage gradually increased to approximately 70% after 24 h. The release of QR was shown to have a biphasic pattern, with a rapid initial release (up to 6 h) and a progressive decline in release rate (up to 24 h.)

### 3.2. QR-Lipo Ameliorates Co-Amox-Induced Liver Damage in Rats

The impact of QR-lipo was then investigated in a rat liver injury model induced by Co-Amox. The most prevalent liver-related enzymes, ALT and AST, and albumin levels were assessed. Following a considerable rise in the Co-Amox rat group due to hepatic damage, our findings showed that QR-lipo therapy returned ALT (Figure 3a) and AST (Figure 3b) to basal levels, such as the healthy control group. Additionally, there was no statistically significant difference in AST and ALT levels between rats given QR-lipo alone and rats in the control group (Figure 3a,b). Finally, there was no distinction in the albumin levels among all groups (Figure 3c).

### 3.3. QR-Lipo Ameliorates Oxidative Damage in Liver Tissue Induced by Co-Amox

To investigate the antioxidative capabilities of QR-lipo, the levels of MDA, GSH, CAT, and TAC in the liver tissues of rats were evaluated. Rats administered Co-Amox showed higher levels of MDA (1.4-fold, Figure 4a), whereas there was a decrease in the levels of GSH (0.76-fold, Figure 4b), CAT (0.71-fold, Figure 4c), and TAC (0.31-fold, Figure 4d) in their liver tissue than in control rats. In addition, combined administration of Co-Amox and QR-lipo significantly reduced the elevated level of MDA by 10% and elevated the decreased levels of GSH, CAT, and TAC compared to the Co-Amox group by 43%, 20%, and 137%, respectively. Rats injected with QR-lipo alone exhibited no significant changes in MDA, GSH, CAT, and TAC levels compared to the untreated control group (Figure 4a–d).

### 3.4. QR-Lipo Reduces Keap1 and Ameliorates Gpx mRNA Expression Levels in Co-Amox-Treated Rats

The influence of the mRNA expression level of genes involved in regulating ROS buildup within cells, such as Keap1 and Gpx, was then evaluated. Compared to healthy control groups, Co-Amox rats displayed an elevation of the mRNA expression of Keap1 (2.6-fold, Figure 5a), which was associated with a 24% decline in the relative mRNA expression of genes related to antioxidant proteins, namely Gpx (Figure 5b). In contrast, treatment with QR-lipo in Co-Amox rats reduced the considerable rise in Keap1 to 0.54-fold when compared to the Co-Amox group (Figure 5a). In addition, QR-lipo improved the suppression of Gpx mRNA gene expression by approximately 4-fold when compared to the Co-Amox group (Figure 5b). At the same time, Gpx and Keap1 mRNA expression levels did not alter significantly between the control group and those treated with QR-lipo alone.

### 3.5. QR-Lipo Ameliorates Inflammatory Gene Expressions in Liver Tissue in Co-Amox-Treated Rats

We aimed to investigate if the regulating mechanism of QR-lipo therapy on ROS formation was related to a change in the hepatic inflammatory genetic signature associated with acute hepatitis caused by oral Co-Amox administration. To address this issue, we analyzed the proinflammatory and anti-inflammatory mRNA expression of livers in Co-Amox rats treated with QR-lipo compared to the group receiving Co-Amox alone. While the liver of the Co-Amox rat group treated with QR-lipo had a significant reduction in mRNA expression of proinflammatory associated genes, including IL-6 (0.49-fold, Figure 6a), IL-1β (0.58-fold, Figure 6b), TNF-α (0.57-fold, Figure 6c), and inducible NO synthase (iNOS) (0.39-fold, Figure 6d), this reduction in the pro-inflammatory signature was associated with a significant decrease in mRNA expression of the NF-κB (0.47-fold, Figure 6e) and an increase in relative mRNA expression of the anti-inflammatory cytokine IL-10 (1.84-fold, Figure 6f) associated genes compared to the liver of rat group receiving Co-Amox alone. In contrast, administering QR-lipo alone did not affect these measured genes, which was similar to the control group.

### 3.6. Microbial Populations of Cecal Contents

Antibiotic oral administration is widely known to disrupt the normal homeostasis of the microbial environment and develop gut dysbiosis. The Co-Amox oral administration caused microbiota dysbiosis to favor colonization by increasing the abundance of *Enterobacteriaceae* (1.91-fold, Figure 7a) and *Clostridium* (3.04-fold, Figure 7e) over the abundance of *Bacteroides* (Figure 7b), *Bifidobacterium* (Figure 7c), and *Lactobacillus* (Figure 7d), which decreased by 16%, 16%, and 50%, respectively. The QR-lipo restored the microbiota population to that of the control, in which *Lactobacillus*, *Bifidobacterium*, and *Bacteroides* colonization increased by 2.7-, 1.5-, and 1.2-fold, respectively, while *Enterobacteriaceae* and *Clostridium* decreased markedly by 0.59- and 0.23-fold, respectively (Figure 7a–e). However, treatment with QR-lipo alone did not affect any of the assessed parameters.

### 3.7. Morphological Changes of the Liver Tissues

The effect of QR-lipo injection on the histopathology of the liver following oral administration of Co-Amox was then evaluated. The livers of rats in the control group and the QR-lipo group displayed normal hepatic parenchyma with no fatty infiltration in hepatocytes or inflammation of the portal area. On the contrary, most of the hepatocytes were swollen in the Co-Amox-treated group, with fuzzy edges and varied sizes. Significant fatty degeneration and portal inflammation were obvious compared to the control group. These pathological findings were improved after QR-lipo treatment. QR-lipo reduced fat degeneration and portal inflammation. Most hepatocytes showed relatively normal histology with few lipid droplets (Figure 8a).

Regarding the liver injury score, the Co-Amox group exhibited a 24.3-fold increase compared to the control group. The Co-Amox and QR-lipo group exhibited an approximately 60% decrease in liver injury score compared to the Co-Amox group, demonstrating that QR-lipo mitigates the damaging effects of Co-Amox on hepatic tissue. The QR-lipo and control groups did not vary significantly (Figure 8b).

### 3.8. Effects of QR-Lipo Treatment on SIRT1 and Nrf2 Protein Expression in Liver Tissues from Co-Amox-Administrated Rats

Immunohistochemical analysis of Nrf2 and SIRT1 in the livers of different groups is shown in Figure 9a–d and Figure 9f–i, respectively. The data obtained from the Co-Amox group showed a decrease in the percent of Nrf2 (Figure 9e) and SIRT1 (Figure 9j) positive areas by approximately 34% and 95%, respectively, in the liver tissues when compared to the control group. QR-lipo treatments consistently increased the percent of Nrf2 (5.04-fold, Figure 9e) and SIRT1 (53.5-fold, Figure 9j) positive areas in the liver tissues of Co-Amox-administrated rats compared to treatment with saline in Co-Amox groups. Additionally, there was no statistically significant difference between the QR-lipo and control groups in the percentage of area in the liver tissues that was positive for Nrf2 and SIRT1.

### 3.9. Molecular Docking Analysis

The computational (in silico) technique has become a popular tool for simulated biological screening during the drug design and discovery phases. This approach assessed the biological functions and predicted the affinities of natural substances, artificial substances, and seminatural molecules to their target sites. Several recent computational chemistry applications have led to a better understanding of the nature of targeted sites and the identification of different compounds as inhibitors or activators. The affinity scores and RMSD values were recorded in Table 2 for both QR and resveratrol as a reference. The binding mechanism of resveratrol (reference for QR) to the SIRT1 target site demonstrated a binding energy of −6.90 kcal/mol. Resveratrol generated two Pi-Alkyl bonds with Pro212 and Ala295, respectively. Resveratrol, however, bonded with Asp292, Asp298 and Lys444 by three hydrogen bonds with bond lengths of 1.70, 1.50, and 1.69 Å (Figure 10a). The binding mechanism of QR to the SIRT-1 target site displayed an energy binding of −9.93 kcal/mol. Three Pi-Alkyl, a Pi-anion, and four strong Pi-Pi stack interactions were formed with Pro212, Ala295, Phe414, and Gln294. Additionally, quercetin interacted with Asp292, Asp298, and Lys444 through three hydrogen bonds with bond lengths of 2.17, 2.00, and 2.13 Å (Figure 10b).

### 3.10. Principal Component Analysis

Principal component analysis (PCA) is a dimension reduction technique that is most beneficial when the data set is large and includes numerous variables [60]. All studied factors were subjected to PCA following loading into the two main components, PC1 and PC2. PC1 and PC2 contributed 88.14 percent to the overall variance. As shown in Figure 11a, all parameters were represented by PC1, which accounts for the biggest share of variation (69.36%), while PC2 contributed substantially less to the variance proportion (18.78%) than PC1. Regarding the PC score plot, the Co-Amox and QR-lipo group demonstrated a negative relationship with PC1, accounting for most of the overall variation. In contrast, the Co-Amox group showed a positive relationship with PC1 (Figure 11b). In the PCA loading plot, GSH, GPx, IL-10, *Bacteroides* count, *Bifidobacterium* count, *Lactobacillus* count, SIRT1, and Nrf2 were significantly correlated with the Co-Amox and QR-lipo group (Figure 11c). While ALT, AST, MDA, Keap1, IL-6, IL-1β, TNF-α, NF-*κ*B, iNOS, *Enterobacteriaceae* count, *Clostridium* count, and liver injury score were found to be coupled with the Co-Amox group (Figure 11c), Figure 11d shows the PCA biplot combination of loading and PC score plots.

## 4. Discussion

Co-Amox is a frequently prescribed antibiotic used to treat a variety of bacterial infections. Despite its extensive track record of therapeutic efficacy, mounting data suggests that it is a leading cause of DILI [8]. This study provides additional evidence that Co-Amox therapy may be coupled with the risk of hepatic injury. In addition, the study revealed the first evidence that QR-lipo can protect against Co-Amox-induced hepatotoxicity. Furthermore, the study elucidated the probable relevance of the SIRT1/Nrf2/NF-κB signaling pathway and the microbiota-liver axis in this pathogenesis.

Obtained results showed that Co-Amox administration in rats increased the levels of ALT and AST, indicating hepatocellular injury. ALT and AST are hepatocyte cytoplasmic enzymes, and their leakage into the bloodstream, particularly ALT, indicates a breakdown of the hepatic membrane. The current evidence of Co-Amox-induced hepatocellular damage in rats is consistent with other previously published research demonstrating large increases in blood levels of AST and ALT in Co-Amox-treated animals [8]. In addition, a considerable amount of previous research demonstrates that Co-Amox toxicity occurs through oxidative stress to produce a detrimental impact on the hepatic tissue that produces higher serum levels of liver damage diagnostic biomarkers [61]. Oxidative stress is initiated by ROS species, such as superoxide anion, hydroxyl radical, and hydrogen peroxide (H_2_O_2_). Furthermore, superoxide radicals can react with nitric oxide, whose production is boosted by an upregulation of iNOS, to produce reactive nitrogen species (RNS), such as nitric oxide and peroxynitrite [1]. Following that, tissue destruction is caused by a range of pathways, including the stimulation of lipid peroxidation, mitochondrial disruption, destruction of DNA, protein nitrification, and apoptosis of cells [62].

Notably, the generation of free radicals in normal cells was tightly regulated by antioxidants. ROS and reactive metabolites of substances such as medications and poisons are eliminated by the body’s superior defense system of enzymatic and nonenzymatic antioxidants. The CAT enzyme, which catalyzes the conversion of hydrogen peroxide to water and oxygen, is one of the first lines of defense against ROS [63]. It has been proven that GSH can participate in several conjugation-based detoxification processes. As a powerful endogenous antioxidant, GSH can scavenge free radicals and decrease peroxides, such as H_2_O_2_. GSH, therefore, provides the cell with numerous defenses against ROS and harmful substances. Nevertheless, the existence of GSH alone is insufficient to safeguard against the cytotoxicity of reactive metabolites, and the presence of GSH-dependent enzymes such as GPx, which contribute to the first and second lines of defense contrary to oxidative stress mediators, is required. GPx is a key selenium-dependent cytosolic enzyme that defends the cell against H_2_O_2_ assault. It interacts with glutathione to produce oxidized glutathione and harmless H_2_O_2_ reduction products. GPx also uses GSH to remove H_2_O_2_ through the production of oxidized glutathione [8]. The current study demonstrated significant damage in hepatic tissue caused by oxidative stress, as evidenced by substantial declines in GSH concentrations, CAT activities, TAC, and GPx gene expression in Co-Amox-treated rats. In contrast, a substantial increase in MDA suggests that lipid peroxidation is fragmenting the membrane lipid content, resulting in a quick and catastrophic collapse of the membrane potential and ion gradients [64]. These outcomes were in line with earlier reports [61,65]. The histopathological analysis confirmed the presence of hepatocellular and cholestatic damage in the form of steatosis, inflammation, cholangitis with bile stasis within the lumen of the bile ducts, and perivascular oedema in the portal area. These results are in accordance with those of Li et al. [66].

ROS can stimulate Kupffer cells, which can initiate the hepatic inflammatory process by producing TNF-α, IL-6, and IL-1β, which support the inflammation process and oxidative stress. In parallel, inflammatory cytokines may boost the accumulation of ROS [67]. Additionally, iNOS-derived NO is a significant factor in fostering an inflammatory response [68] by upregulating NF-κB, a key transcription factor in inflammatory reactions [69]. The data obtained demonstrated that Co-Amox-treated rats exhibited an increase in NF-κB gene expression, which is crucial for regulating the release of inflammatory cytokines and may also increase the gene expression of TNF-α, IL-6, and IL-1β mRNAs. These findings are congruent with those of Yu Guo et al. [70], who observed an increase in these markers in chickens with amoxicillin/potassium clavulanate-induced liver damage.

QR, a natural flavonoid, is thought to be a powerful antioxidant [29]. The molecular structure of QR, which consists of an active oxygen group, phenolic hydroxyls, and a 2,3-unsaturated double bond, is associated with a significant antioxidant ability that enables it to take oxygen free radicals and produce metal chelation compounds [71]. QR is also able to lower iNOS and consequently prevent the excessive production of NO [72]. Additionally, it guards against DNA damage, protein oxidation, and lipid peroxidation [71]. However, QR’s high hydrophobicity (log P 1.8) [73] and poor water solubility (5 µg/mL) hinder its best use. The formulation of QR in liposomes could enhance QR’s dissolution, solubility, and bioavailability [74]. The developed liposomal formulation of QR was found to enhance QR loading, control QR release, and improve its biological activity.

The acquired results suggested that pretreatment with QR-lipo considerably decreased Co-Amox-induced hepatotoxicity by lowering oxidative stress, improving liver histology, and restoring ALT and AST levels. The ROS-inhibiting properties of QR nanoliposomes may be responsible for these outcomes, which include raising CAT activity, raising GSH and GPx gene expression levels, and lowering MDA and iNOS gene expression. These results are consistent with those of other researchers who investigated the hepatoprotective effects of QR-lipo against alcoholic fatty liver and thioacetamide-induced hepatotoxicity in rats by restoring the depleted levels of GSH and activities of CAT [71,75]. Another study demonstrated QR-lipo’s potential to mitigate the acute liver damage caused by paracetamol intoxication in rats by lowering iNOS and RNS [76], which is consistent with the obtained results.

Alongside the antioxidant properties, according to our findings, the pretreatment with QR-lipo activated the gene expression of the most important anti-inflammatory cytokine, IL-10, which can suppress proinflammatory responses [77]. It also inhibited gene expression of IL-6, IL-1β, TNF-α, and NF-κB mRNAs. Similarly, other researchers reported that QR could reduce inflammation caused by nonalcoholic fatty liver disease [75]. Prior studies have demonstrated QR’s antioxidant activity, which is most likely due to its interference with the NF-κB pathway in rat models of thioacetamide-induced acute liver toxicity [78] and doxorubicin-induced liver toxicity [79].

To further explain the route, the immunohistochemistry of SIRT1 and Nrf2 was examined. SIRT1 is believed to be a critical regulator in the activation of genes related to oxidative stress or inflammation, according to credible data [80]. In addition, it has been observed that SIRT1 stimulates the Nrf2 pathway by lowering ROS generation. Nrf2 is a critical transcription factor for preventing oxidative damage to cells. Under typical conditions, Nrf2 is inactive when coupled with Keap1 [81]. While excessive oxidative stress results in Keap1 dissociation, it also consumes a lot of Nrf2 and disturbs homeostasis [80]. Likewise, it has been suggested that Nrf2 and NF-κB have an adversarial relationship. Therefore, SIRT1 or Nrf2 activation may be a significant target for reducing inflammation in liver diseases [82]. The data obtained indicated that pretreatment with QR-lipo improved the ability to battle oxidative stress and inflammation by increasing the expression of SIRT1 and Nrf2, reversing their depletion, and reducing the elevated levels of Keap1 induced by Co-Amox treatment. These findings are comparable with those of other researchers who found a significant reduction in SIRT1 and Nrf2 in acetaminophen-induced hepatotoxicity and HFD/STZ-induced diabetic liver disease, respectively [80,82]. Simultaneously, another team showed a protective effect of QR against nickel and hyperthyroidism-induced hepatic damage via the Nrf2 signaling pathway [83,84]. Furthermore, Zhang et al. [85] reported that pretreatment with QR protects against isoniazid-induced liver injury by regulating the SIRT1 pathway. A molecular docking that further supported the effect of QR-lipo on SIRT1 revealed that QR has the potential to act as an activator at the SIRT1 target site, potentially modulating its activity similarly to resveratrol [86].

Previous studies have addressed the effect of extensive use of antibiotics on the disturbance of the microbiota ecosystem balance through not only the reduction in microbiota diversity but also the reduction of beneficial bacteria colonization efficiency and, consequently, an increase in harmful ones [22,87]. There is a strong link between microbiota dysbiosis and metabolic diseases, including obesity, diabetes [88,89], and hepatic disease progression [14]. For instance, Henao-Mejia and his colleagues demonstrated that the progression of liver inflammation in nonacholic fatty liver disease to chronic hepatic inflammation associated with cirrhosis was attributed in part to disturbance in the gut microbiota leading to exacerbated hepatic injury through stimulation of TNF-α and NF-κB signaling pathways upon translocation of their product through the hepatic portal vein, as indicated by an increase in toll-like receptor agonist in the portal circulation of inflammasome deficient mice and cohoused cagemates wild type compared to single housed mice [16]. Recently, global attention has been focused on studying the effects of natural substances to be used either as food supplements or antibiotic alternatives for combating the problems of gut dysbiosis and bacterial resistance. It has been known that feeding QR in diet supplements has a prebiotic effect on the modulation of gut microbiota dysbiosis induced either by antibiotics, a high-fat diet, as well as chemical diabetic-induced mice [87,89,90]. This study investigated the prebiotic effect of QR-lipo intraperitoneal injection on microbiota dysbiosis induced by Co-Amox oral administration. The findings obtained indicated that Co-Amox induced microbiota disturbance by favoring the colonization of *Enterobacteriaceae* and *Clostridium* species over the abundance of beneficial bacteria, including *Lactobacillus*, *Bifidobacterium,* and *Bacteroides*, as indicated by the CFU of the bacteria in the cecal contents. However, pretreatment of Co-Amox-treated rats with QR-lipo showed a prebiotic effect on gut microbiota, as indicated by a significant increase in abundance of *Bifidobacterium*, *Bacteroides,* and *Lactobacillus* at the expense of *Clostridium* and *Enterobacteriaceae* within the cecal contents compared to the untreated group. Shi et al. [22] demonstrated that intragastric administration of an antibiotic cocktail containing vancomycin, neomycin sulfate, metronidazole, and ampicillin caused a significant reduction in total bacterial abundance, the diversity of the bacterial community, and a decrease in *Bacteroides* population in mice. The present study demonstrated that using QR as a diet supplement for 10 days following antibiotic cocktail treatment increased gut *Bacteroides* abundance, similar to healthy controls, which is in alignment with the obtained results [22]. Collectively, the results suggested the prebiotic effect of QR-lipo on gut microbial ecology and its protective effect against antibiotic-induced microbiota dysbiosis.

According to the obtained data, the pretreatment of QR-lipo with Co-Amox in rats demonstrated a considerable increase in the colonization of *Bifidobacterium* and *Lactobacillus*. Both *Bifidobacterium* and *Lactobacillus* were able to promote the growth of butyrate-producing bacteria [91,92]. This would suggest the role of QR in augmenting butyrate production in gut ecology. In addition, short-chain fatty acids, including butyrate, play an important role in maintaining intestinal homeostasis through the stimulation of Tregs and the antimicrobial activity of intestinal macrophages [93]. This may partially explain the effect of QR-lipo pretreatment on the ameliorative phenotype observed in liver inflammation and steatosis severity in Co-Amox-treated rats. However, further transcriptomic and metabolomic analysis would be required for the investigation of QR-lipo’s effect on amelioration of liver injury. The study also did not address the possibility that the effect of QR-lipo on inflammatory mediators may be associated with changes in liver macrophage phenotypes, which would be recommended in the future. In addition, the study administered QR-lipo by intraperitoneal injection, which is regarded as an invasive route of drug delivery. Therefore, one of our future perspectives is to develop a noninvasive method for the delivery of QR and other promising phytoceuticals.

## 5. Conclusions

In this work, QR-lipo was developed and successfully employed to mitigate antibiotic-induced liver injury. QR-lipo pretreatment activated the SIRT1/Nrf2/NF-κB signaling pathway to improve antioxidant and anti-inflammatory effects in Co-Amox-treated rats. Furthermore, QR-lipo inhibited antibiotic-induced microbiota dysbiosis. The study suggested that QR-lipo might ameliorate Co-Amox-induced hepatic insults through multifactorial protective pathways.

## Figures and Tables

**Figure 1 antioxidants-12-01487-f001:**
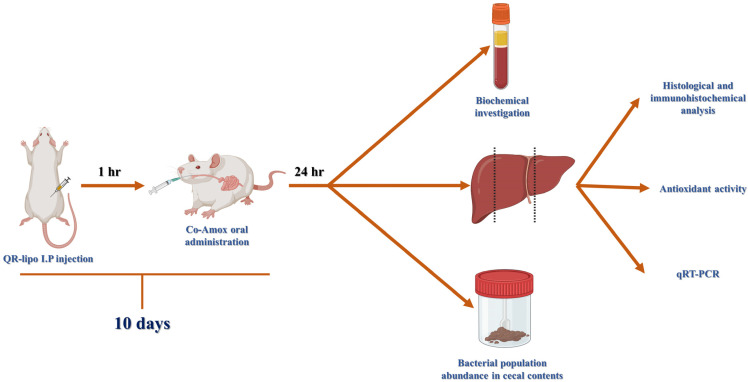
Schematic diagram summarizing the experimental design.

**Figure 2 antioxidants-12-01487-f002:**
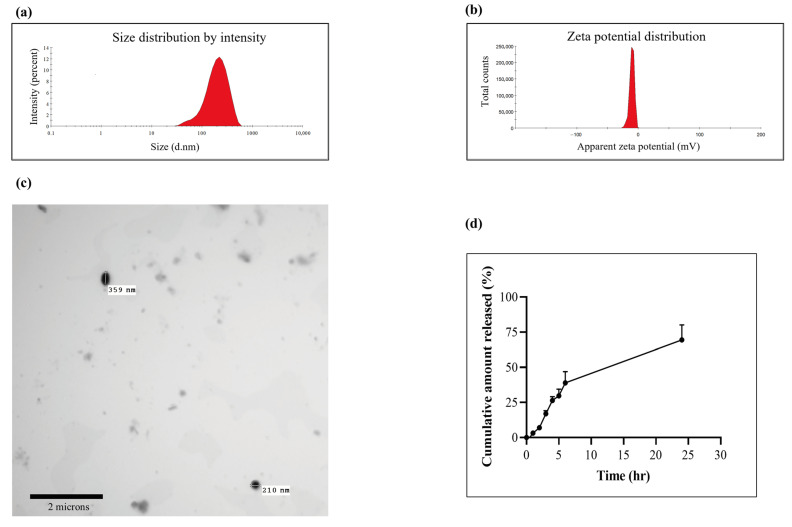
Characterization of QR-Lip for (**a**) particle size, (**b**) zeta potential, (**c**) transmission electron microscopy, and (**d**) in vitro QR release from QR-lipo. The ethanol injection technique was successful in the preparation of QR-lipo. The characterization of QR-lipo confirmed that it had a spherical nature, adequate zeta potential, and nano-size dimensions. In vitro release studies showed enhanced drug release upon formulation into nanoliposomes.

**Figure 3 antioxidants-12-01487-f003:**
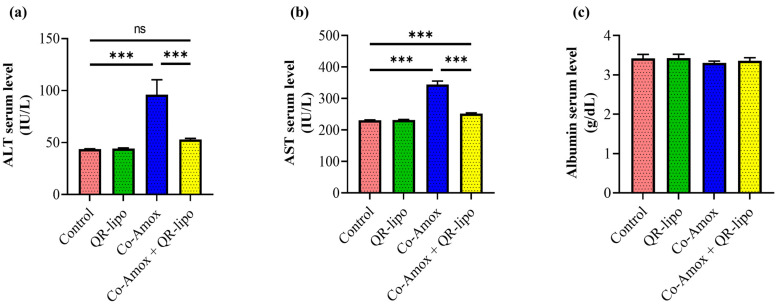
Effect of QR-lipo pretreatment on liver function tests in Co-Amox treated rats. The serum levels of (**a**) ALT, (**b**) AST, and (**c**) albumin were quantified for the different experimental groups. Data are expressed as the mean ± SD (*n* = 7). ns *p* > 0.05 and *** *p* < 0.001.

**Figure 4 antioxidants-12-01487-f004:**
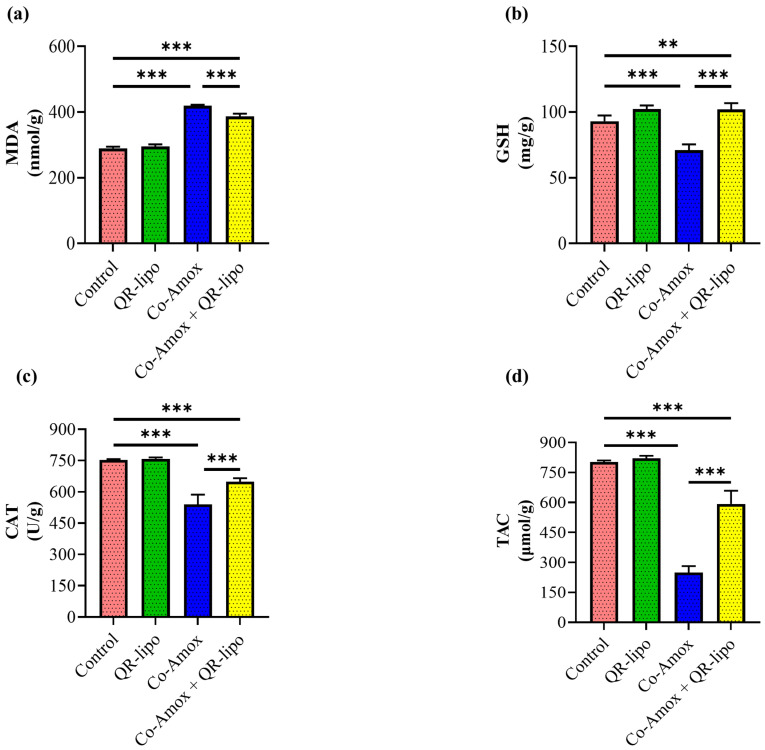
Effect of QR-lipo pretreatment on liver antioxidant status in Co-Amox-treated rats. The liver antioxidant status was determined via the quantification of (**a**) MDA, (**b**) GSH, (**c**) CAT, and (**d**) TAC. Data are expressed as the mean ± SD (*n* = 7). ** *p* < 0.01, and *** *p* < 0.001.

**Figure 5 antioxidants-12-01487-f005:**
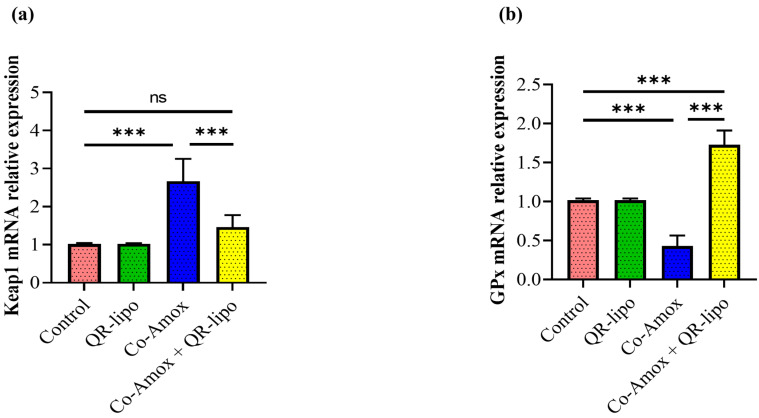
Effect of QR-lipo pretreatment on the mRNA expression level of ROS regulation genes within hepatocytes in Co-Amox-treated mice. The level of expressed mRNA for (**a**) keap1 and (**b**) GPx genes were detected using qRT-PCR. Data are expressed as the mean ± SD (*n* = 7). ns *p* > 0.05, *** *p* < 0.001.

**Figure 6 antioxidants-12-01487-f006:**
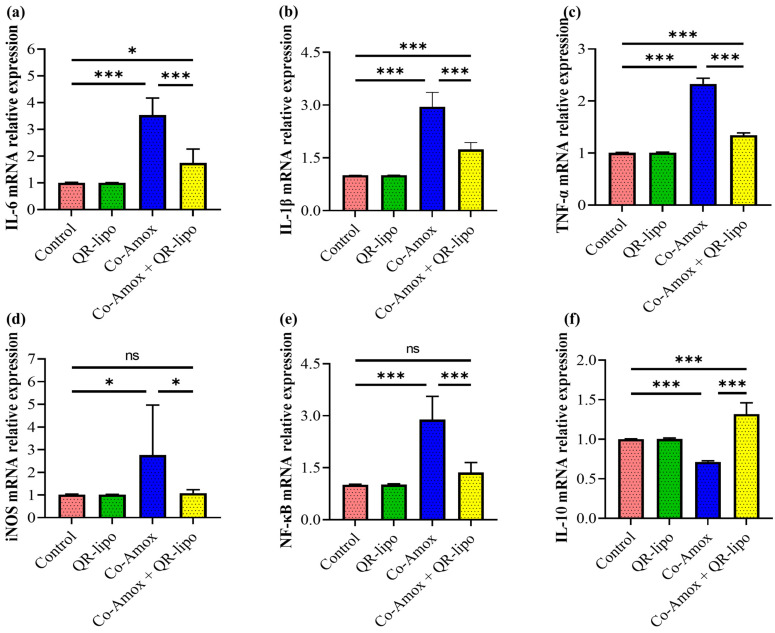
Effect of QR-lipo pretreatment on mRNA level of inflammatory-related genes in liver tissues of Co-Amox-treated rats. The relative mRNA expression level of proinflammatory mediators, including (**a**) IL-6, (**b**) IL-1β, (**c**) TNF-α, (**d**) iNOS, and (**e**) Nf-κB, and anti-inflammatory mediator (**f**) IL-10 was quantified using RT-PCR for the different experimental groups. Data are expressed as the mean ± SD (*n* = 7). ns *p* > 0.05, * *p* < 0.05, and *** *p* < 0.001.

**Figure 7 antioxidants-12-01487-f007:**
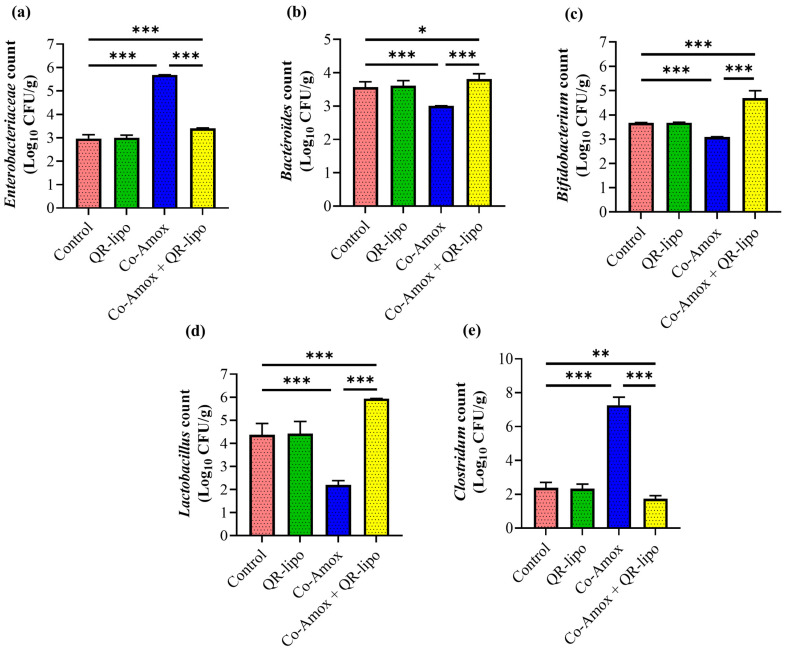
Effect of QR-lipo pretreatment on microbial populations in cecal contents of Co-Amox-treated rats. The Graph represented the counts of bacterial species expressed as log10 CFU/g of the sample; (**a**) *Enterobacteriaceae*, (**b**) *Bacteroides*, (**c**) *Bifidobacterium*, (**d**) *Lactobacillus*, and (**e**) *Clostridium*. Data are obtained from rat experiments and expressed as the mean ± SD (*n* = 7). * *p* < 0.05, ** *p* < 0.01, and *** *p* < 0.001.

**Figure 8 antioxidants-12-01487-f008:**
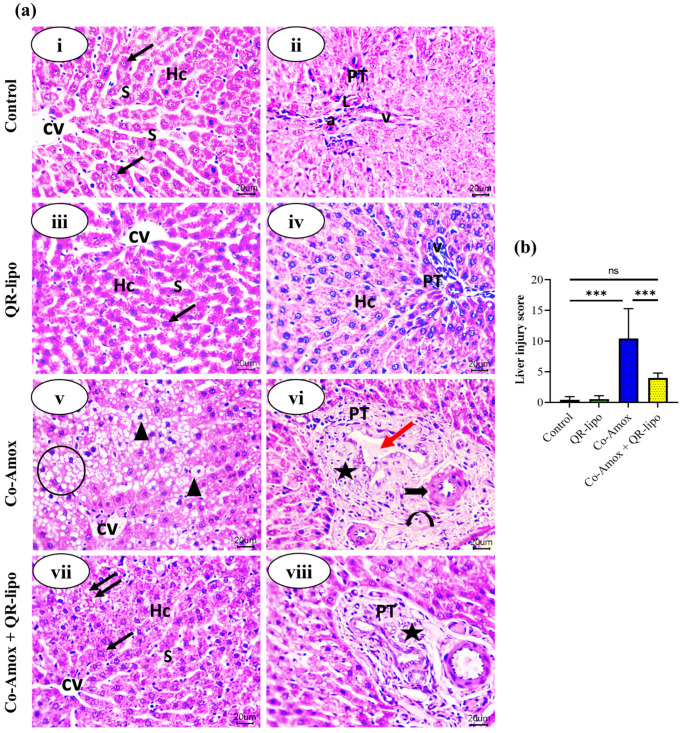
Effect of QR-lipo pretreatment on histological changes in liver tissue in Co-Amox-treated rats. (**a**) Photomicrographs of the liver tissue stained with hematoxylin-eosin. (**i**,**ii**) Control group: showing normal hepatic architecture. A small central vein in the center of the hepatic lobule (CV). Plates of hepatocytes (Hc) are arranged radially around the small central vein. Each hepatocyte has an acidophilic cytoplasm and central vesicular nucleus (arrow). Hepatic sinusoids (s) run between the cords of hepatocytes. The portal tract (PT) at the periphery of each lobule is composed of an area of connective tissue with a small branch of the hepatic artery (a), portal vein (v), and lymphatic vessel (L). (**iii**,**iv**) QR-lipo group: nearly the same histological structure as the control group. (**v**,**vi**) Co-Amox group: shows lost normal hepatic architecture. Widespread fatty degeneration appeared as excessively small intracytoplasmic droplets with nuclear pyknosis (arrowheads). Some droplets coalesce against each other, forming cellular ballooning (circle). Chronic cholangitis in the portal tract (star) with dilated portal vein and bile stasis (red arrow) is also found. Perivascular oedema within the portal area (curved arrow) and increased thickness of the hepatic artery (black thick arrow) can be seen. (**vii**,**viii**) Co-Amox and QR-lipo group: nearly normal anastomosing plates of hepatocytes (Hc), central vein (CV), and blood sinusoids (s). Most hepatocytes appear normal with a central vesicular nucleus and eosinophilic cytoplasm (arrow), except for a few focal areas of vacuolated hepatocytes with small lipid droplets can be seen (double arrows). Improved portal area with mild inflammatory reaction (star) can be seen. (H and E; Scale bar; 20 μm). The magnification power is set at 400×. (**b**) Liver injury score. ns *p* > 0.05 and *** *p* < 0.001.

**Figure 9 antioxidants-12-01487-f009:**
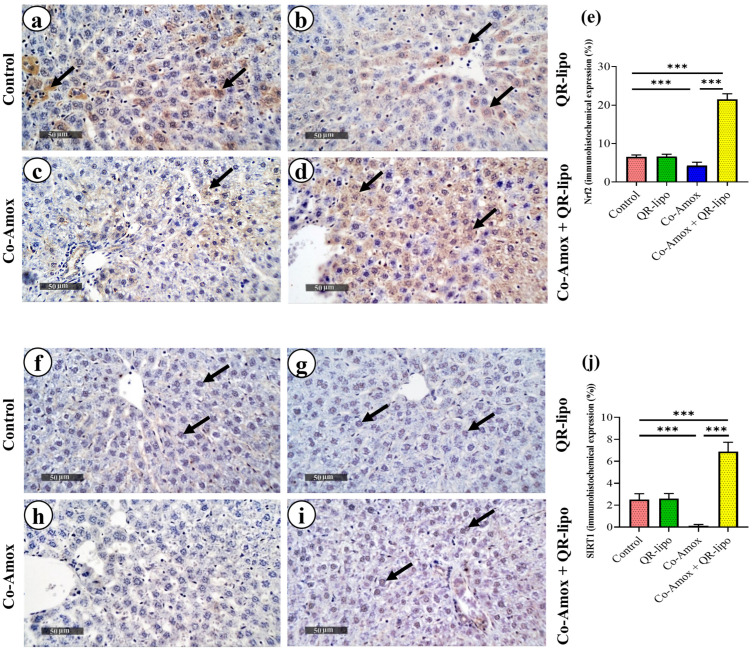
Photomicrographs of sections in the liver of rats stained with (**a**–**d**) anti-Nrf2 antibody and (**f**–**i**) anti-SIRT1 antibody. (**a**–**d**) Expression of Nrf2 mainly in the hepatocyte cytoplasm of (a) control group: moderate expression and brown stained cytoplasm indicating positive reaction (arrows). (**b**) QR-Lipo group: similar to the control group. (**c**) Co-Amox group: downregulated Nrf2 expression in hepatocytes cytoplasm and most cells are showing weak reactions. (**d**) Co-Amox and QR-lipo: high increase in Nrf2 expression in the cytoplasm of the hepatocytes (scale bar; 50 μm and magnification power; 400×). (**f**–**i**) Expression of SIRT1 in the hepatocyte nuclei of (**f**) control group: moderate expression and brown-stained nuclei indicate positive reaction (arrows). (**g**) QR-lipo group: moderate expression similar to the control group. (**h**) Co-Amox group: weak expression of SIRT1 in hepatocyte nuclei. (**i**) Co-Amox and QR-Lipo group: strong increase in SIRT1 expression (Scale bar; 50 μm and magnification power; 400×). (**e**,**j**) Mean area percentages of Nrf2 and SIRT1 immune expression, respectively, for the different experimental groups. *** *p* < 0.001.

**Figure 10 antioxidants-12-01487-f010:**
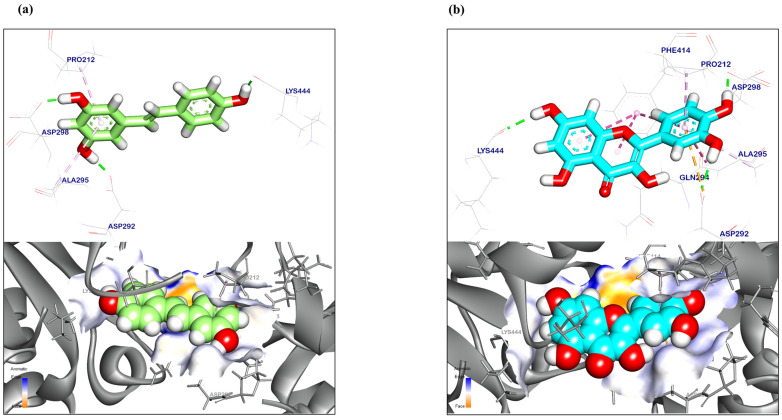
Three-dimensional orientation and surface mapping against SIRT-1 target site of (**a**) Resveratrol (reference) and (**b**) QR.

**Figure 11 antioxidants-12-01487-f011:**
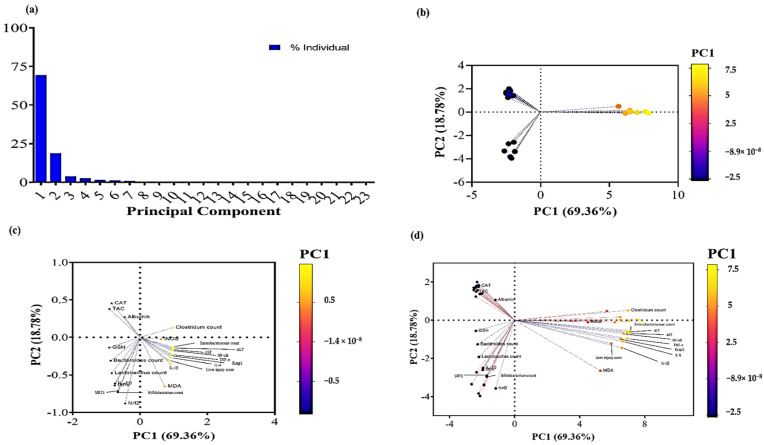
Principal component analysis of all variables in the study. Principal component analysis for all variables was conducted to summarize the data. (**a**) Contribution of principal components to the total variance. (**b**) PC scores plot showing the different groups’ dimensions based on the PC1 scale. (**c**) PC loading plot showing the different variables’ dimensions based on the PC1 scale. (**d**) PC biplot combining the PC scores and PC loading plots.

**Table 1 antioxidants-12-01487-t001:** Primer sequences used for real-time quantitative reverse transcriptase-polymerase chain reaction (qRT-PCR).

Genes	Primer	Sequence (5′-3′)	Accession Number/References
Keap1	Forward	TGGGCGTGGCAGTGCTCAAC	NM_057152/[44]
	Reverse	GCCCATCGTAGCCTCCTGCG	
Gpx	Forward	GGTGTTCCAGTGCGCAGAT	X12367.1/[45]
	Reverse	AGGGCTTCTATATCGGGTTCGA	
IL-6	Forward	TCCTACCCCAACTTCCAATGCTC	NM_012589.2/[46]
	Reverse	TTGGATGGTCTTGGTCCTTAGCC	
IL-1β	Forward	CACCTCTCAAGCAGAGCACAG	NM_031512.2/[46]
	Reverse	GGGTTCCATGGTGAAGTCAAC	
TNF-α	Forward	AAATGGGCTCCCTCTCATCAGTTC	L19123.1/[46]
	Reverse	TCTGCTTGGTGGTTTGCTACGAC	
IL-10	Forward	GCAGGACTTTAAGGGTTACTTGG	L02926.1/[47]
	Reverse	GGGGAGAAATCGATGACAGC	
NF-*κ*B	Forward	AATTGCCCCGGCAT	XM_342346.4/[48]
	Reverse	TCCCGTAACCGCGTA	
iNOS	Forward	CACCACCCTCCTTGTTCAAC	NM_012611/[49]
	Reverse	CAATCCACAACTCGCTCCAA	
*Enterobacteriaceae*	Forward	CATTGACGTTACCCGCAGAAGAAGC	CU928145/[50]
	Reverse	CTCTACGAGACTCAAGCTTGC	
*Bacteroides*	Forward	GAGAGGAAGGTCCCCCAC	NC_003228/[51]
	Reverse	CGCTACTTGGCTGGTTCAG	
*Bifidobacterium*	Forward	CTC CTG GAA ACG GGT GG	CP001213/[52]
	Reverse	GGT GTT CTT CCC GAT ATC TAC A	
*Lactobacillus*	Forward	AGCAGTAGGGAATCTTCCA	NC_015213/[53]
	Reverse	CACCGCTACACATGGAG	
*Clostridium*	Forward	AAAGGAAGATTAATACCGCATAA	KF929215/[54]
	Reverse	ATCTTGCGACCGTACTCCCC	
β-actin	Forward	CCTGCTTGCTGATCCACA	V01217.1/[55]
	Reverse	CTGACCGAGCGTGGCTAG	

**Table 2 antioxidants-12-01487-t002:** DG, RMSD, interactions (kcal/mol) of resveratrol (reference), and QR against the SIRT1 target site.

Targets	Tested Compounds	RMSD Value (Å)	Docking (Affinity) Score (kcal/mol)	Interactions	
				H. B	Pi-Interaction
SIRT-1	Resveratrol (Reference)	0.75	−6.90	3	2
	QR	1.220	−9.93	3	7

## Data Availability

Not applicable.

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
