# Peer review of "The Potential Effects of Quercetin-Loaded Nanoliposomes on Amoxicillin/Clavulanate-Induced Hepatic Damage: Targeting the SIRT1/Nrf2/NF-κB Signaling Pathway and Microbiota Modulation"

_antioxidants, 2023, doi:10.3390/antiox12081487_

Round 1

Reviewer 1 Report

Abstract

The abstract must be rewritten. It's confusing, needs to be more structured with inclusion of rat line, techniques used. In addition, you must define the role of SIRT1 and Nrf2.

Introduction

The introduction is well written. However, a paragraph on resveratrol and its relationship with Sirtuin-1 is missing.

Material and Methods

Animals and experimental design

-Please specify the sex of animals

-Include a diagram of your protocol

-Justify the dose you used

Sampling

When will the blood be collected, after or before the sacrifice, and what is the last treatment?

Results and figures

-Please under each figure include the p values significance

-In the analyze of the results, include values or percentage. For me, indicating an increase or decrease is not enough.

-Figure 7 and 8: The pictures should be much larger than the graph

Discussion

-The discussion is well written.

-Figure 11: a summary of effects is required in the legend

-Please indicate the limitation of our study

Author Response

Alsalamoo aleekom,

Thank you for comments.

Regards

Reviewer 2 Report

Manuscript antioxidants-2494052, entitled “The Potential Effects of Quercetin-loaded Nanoliposomes on Amoxicillin/clavulanate Induced Hepatic Damage: Targeting SIRT1/Nrf2/NF-κB Signaling Pathway and Microbiota Modulation.”

Recommendation:       The above paper is not suitable for publication in its present form.

General comment

This research article provides useful information about the effects of quercetin-loaded nanoliposomes on amoxicillin/clavulanate induced hepatic damage, as indicated by the SIRT1/Nrf2/NF-κB signaling pathway and microbiota modulation. Although, it is in general appropriately organized and written, there are some points that should be corrected or clarified.

L33-34: “…hepatotoxicity. The aim of this study was therefore to investigate…”

L37: “elevated levels”?

L38: “…lipid peroxidation value and…”

L42: “by” instead of “and”

L43-44: “However, QR-lipo effectively ameliorated CoA-induced…”

L48: “maintenance of” instead of “maintaining”

L54: “reported” instead of “seen”

L56: “accused” instead of “to blame”

L58: “…is the most often associated with DILI…” Something is missing? Compound?

L65-66: Please rephrase

L71: “circumstances” instead of “instances”

L74: “scavengers become” instead of “scavenging becomes”

L76: “actor”?

L76: “development” instead of “research”

L80: “…is downregulated as an effect of hepatocellular…”

L105: “impairs” instead of “impairing”

L108: “an alternative” instead of “demanding”

L115: “possess” instead of “have”

L116: “properties” instead of “effects”

L142: “provided with” instead of “given”

L149: “was observed”?

L151: “was” instead of “is”

L161: “…was also examined…”

L166-167: “…50–60 rpm, respectively. Two mL of the dissolving medium were collected at regular…”

L171: “allocated” instead of “separated”

L173-174: “The liver injury was induced by CoA oral suspension…”

L176-179: Please rephrase

L187-188: Please delete “However”

L232: Where is the ratio?

L276-277: “…for each tissue section per sample.”

L285-287: “The key amino acids (Lys444, 285 Asp292, Ala295, Asp298 and Pro212), which are responsible for the activation of the SIRT1 domain by activators attachment (Resveratrol) were identified [58].”

L305: Please rephrase

L309: “shown” instead of “seen”

L329-330: “…group (Figures 2a-b). Finally, there was no distinction in the albumin levels among all groups (Figure 2c).”

L343: Please delete “While”

L344: “compared” instead of “relative”

L406: Please delete “Next”

L407: “…of CoA was then evaluated.”

L545: “H2O”?

L560: “are in accordance with” instead of “parallel”

L575: “is associated with” instead of “gives it”

L577: “…prevents the excessive production of NO [72].”

L588: “…consistent with these of other researchers…”

L592: “…which is consistent with…”

L605: “…pathway, by lowering ROS generation.”

L619-620: Please remove “[85]” after the name of the authors”

L623: Please delete “been”

L635: “…attention is focused on studying…”

L647: Please delete “abundance”

L648: “…group. Shi et al. [22] demonstrated…”

L651: “…in mice. The present study…”

Minor editing of English language required

Author Response

Alsalamoo aleekom

Dear Dr,

Thank you.

Regards

Reviewer 3 Report

Title: 

The Potential Effects of Quercetin-loaded Nanoliposomes on Amoxicillin/clavulanate Induced Hepatic Damage: Targeting SIRT1/Nrf2/NF-kB Signaling Pathway and Microbiota Modulation. 

Authors: 

El-Emam M.M.A., Mostafa M., Farag A.A., Youssef H.S., El-Demerdash A.S., Bayoumi H., Gebba M.A., El-Halawani S.M., Saleh A.M., Badr A., El Sayed S. 

Manuscript ID: antioxidants-2494052

Objective: 

This manuscript presents the numerous protective effects of quercetin nanoliposomes (QR-lipo) against liver damage induced by amoxicillin/clavulanate, an antibiotic. QR-lipo has an anti-inflammatory, antioxidant effect - against oxidative stress, whereby these effects have been ascertained both at the molecular and genetic levels. Furthermore, a positive impact on the microbiome could also be demonstrated.

Points of criticism:

The abbreviation (CoA) for amoxicillin/clavulanate should be changed because it could be confused with coenzyme A, for example, in literature searches.

Page 3 – line 126:

… that are poorly water-soluble including QR are now more bioavailable [32].” 

What do the authors mean by "now"? Does it mean that liposomes make it more bioavailable? This should be specified.

Page 4 – line 149:

Ten "ml"... should be written out the first time and recognisable in round brackets as an abbreviation for the rest of the manuscript.

Page 4 – line 167:

… was “taken out" should be replaced by obtained.

Page 5 – line 220:

(GmbH, Qiagen, Germany) should read as indicated on line 238 – (Qiagen, Hilden, Germany).

Page 7 – line 278:

(Leica Microsystems GmbH, Germany) – Ltd. should substitute GmbH, and the corresponding city should be mentioned.

Figure 3, 7, 8:

The presentation of Fig. 3a, 3b, 7b, and 8j should be improved. The different scaling of the y-axis is confusing and, apart from Fig 8e, unnecessary.

Figures 7 & 8:

The magnification of the photomicrographs should be mentioned.

Page 18 - line 528:

According to their results, the authors refer to further animal studies - ref. 8 and 61. In the case of Ref. 61, the side effect is described in a 22-year-old female. This reference is, therefore, inappropriate and should be removed.

Page 19 – line 545:

H2O should read H2O2.

Page 19 – line 551:

Insert “harmless” in front of ”… H2O2 reduction product”.

Page 19 – line 552:

“The current study demonstrated significant damage …”. The authors presumably mean liver damage – which should be mentioned in this sentence.

References:

The reference section should be improved thoroughly, i.e., correct abbreviations of the Christian names of the authors, journal names (sometimes abbreviated, sometimes at full length), volume and page numbers and congruency with mentioning of DOI. Out of the first 20 references, solely Ref. 2, 4, 5, 9, 13, 16, and 17 were correctly cited. Severe mistakes were observed for Ref 3 – Van Raaphorst is not an author of this manuscript; Ref. 6 - Author names and journal name and Ref. 20 (page 23 – line 751) mistake after Broiler Chickens.

Author Response

Alsalamoo aleekom,

Dear Dr.

Thank you.

Regards

Round 2

Reviewer 1 Report

The manuscript was greatly improved and the authors' responses were satisfactory. In my opinion, the manuscript can now be accepted for publication.